# What Do Deep Saliency Models Learn about Visual Attention?

**Shi Chen**      **Ming Jiang**      **Qi Zhao**
Department of Computer Science and Engineering,
University of Minnesota
{chen4595, mjiang, qzhao}@umn.edu

## Abstract

In recent years, deep saliency models have made significant progress in predicting human visual attention. However, the mechanisms behind their success remain largely unexplained due to the opaque nature of deep neural networks. In this paper, we present a novel analytic framework that sheds light on the implicit features learned by saliency models and provides principled interpretation and quantification of their contributions to saliency prediction. Our approach decomposes these implicit features into interpretable bases that are explicitly aligned with semantic attributes and reformulates saliency prediction as a weighted combination of probability maps connecting the bases and saliency. By applying our framework, we conduct extensive analyses from various perspectives, including the positive and negative weights of semantics, the impact of training data and architectural designs, the progressive influences of fine-tuning, and common failure patterns of state-of-the-art deep saliency models. Additionally, we demonstrate the effectiveness of our framework by exploring visual attention characteristics in various application scenarios, such as the atypical attention of people with autism spectrum disorder, attention to emotion-eliciting stimuli, and attention evolution over time. Our code is publicly available at `https://github.com/szzexpoi/saliency_analysis`.

## 1 Introduction

Attention deployment is a complex and fundamental process that enables humans to selectively attend to important sensory data in the visual environment. Scientists have been fascinated by the question of what drives human visual attention for decades. Understanding the mechanisms of visual attention not only sheds light on the human visual system but also helps computational methods to localize critical sensory inputs more efficiently.

One approach to predicting human attention is through saliency models, which have received considerable research attention. A saliency model predicts the most visually important regions in an image that are likely to capture attention. Earlier models follow a feature integration approach [1, 2, 3, 4] and extract low-level features (*e.g.,* colors, intensity, orientations) or higher-level features (*e.g.,* objects, semantics) from the input image to infer saliency [5, 6, 7]. While these models show initial success, their performance is limited by the difficulty of engineering relevant visual features. In contrast, recent saliency models [8, 9, 10, 11] follow a data-driven approach, leveraging large datasets [12] and deep neural networks [13, 14, 15] to learn discriminative features. These models achieve human-level performance on several saliency benchmarks [12, 16, 17], thanks to their accurate detection of important objects and high-level semantics [18, 19]. However, due to the lack of transparency, it is still unclear what semantic features these models have captured to predict visual saliency.

To understand how deep neural networks predict visual saliency, in this paper, we develop a principled analytic framework and address several key questions through comprehensive analyses:

- How do deep saliency models differentiate salient and non-salient semantics?
- How do data and model designs affect semantic weights in saliency prediction?
- How does fine-tuning saliency models affect semantic weights?
- Can deep saliency models capture characteristics of human attention?
- What is missing to close the gap between saliency models and human attention?

Our method connects implicit features learned by deep saliency models to interpretable semantic attributes and quantifies their impact with a probabilistic method. It factorizes the features into trainable bases, and reformulates the saliency inference as a weighted combination of probability maps, with each map indicating the presence of a basis. By measuring the alignment between the bases and fine-grained semantic attributes (*e.g.,* concepts in Visual Genome dataset [20]), it quantifies the relationships between diverse semantics and saliency. This unique capability enables us to identify the impact of various key factors on saliency prediction, including training datasets, model designs, and fine-tuning. It can also identify common failure patterns with state-of-the-art saliency models, such as SALICON [9], DINet [11], and TranSalNet [10]. Beyond general saliency prediction, the framework also shows promise in analyzing fine-grained attention preferences in specific application contexts, such as attention in people with autism spectrum disorder, attention to emotional-eliciting stimuli, and attention evolution over time. In sum, our method offers an interpretable interface that enables researchers to better understand the relationships between visual semantics and saliency prediction, as well as a tool for analyzing the performance of deep saliency models in various applications.

## 2   Related Works

Our work is most related to previous studies on visual saliency prediction, which make progress in both data collection and computational modeling.

**Saliency datasets.** With the overarching goal of understanding human visual perception, considerable efforts have been placed on constructing saliency datasets with diverse stimuli. The pioneering study [16] proposes an eye-tracking dataset with naturalistic images, which later becomes a popular online benchmark [21]. Several subsequent datasets characterize visual saliency into finer categories based on visual scenes [22], visual semantics [17], sentiments [23, 24], or temporal dynamics [25], to study the impact of different experimental factors on attention. To overcome the difficulties of tracking eye movements, Jiang *et al.* [12] leverage crowd-sourcing techniques and use mouse-tracking as an approximation for eye gaze, which results in currently the largest saliency dataset. Recent works also consider broader ranges of visual stimuli, including videos [26, 27], graphical designs [28, 29], web pages [30, 31], crowds [32], driving scenes [33] and immersive environments [34, 35]. In this study, we focus on visual saliency for naturalistic images, which serves as the foundation of saliency prediction studies.

**Saliency models.** Prior works have developed saliency prediction models to quantitatively study human attention. Inspired by seminal works on saliency modeling [5, 6, 36], early saliency models [1, 2, 3, 4, 37] typically adopt a bottom-up approach, integrating handcrafted features (*e.g.,* colors, intensity, and orientations). On the other hand, recent approaches take a different route and leverage deep neural networks [13, 14, 15, 38] to automatically learn features and predict saliency. Vig *et al.* [39] is one of the first attempts to utilize convolutional neural networks (CNNs) for saliency prediction. Huang *et al.* [9] consider features learned from multi-scale inputs to model the coarse-to-fine semantics. Kruthiventi *et al.* [40] leverage convolutional layers with diverse kernel sizes to capture multi-scale features and incorporate positional biases with location-dependent convolution. Kümmerer *et al.* [41] demonstrate the usefulness of features of visual objects in saliency prediction. Cornia *et al.* [8] develop a recurrent neural network to iteratively refine features for saliency prediction. Yang *et al.* [11] improve saliency prediction with dilated convolution to capture information from broader regions. Lou *et al.* [10] study the usefulness of self-attention [15] for saliency prediction. Instead of building new models, several works [18, 42, 43, 44, 45, 46] study the behaviors of models. By analyzing predictions on different categories of stimuli, they identify key factors behind the successes and failures of existing saliency models (*e.g.,* accurate detection of semantic objects, incorporation of low- and high-level features, contrast between diverse visual cues, detection of Odd-One-Out target, and etc.), and propose directions for improvements. A recent work [19] also

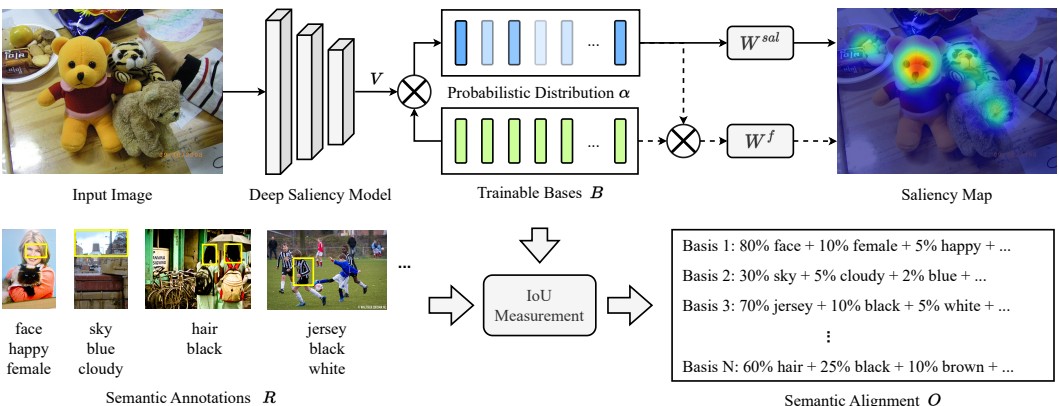

Figure 1: Illustration of our method. It factorizes implicit features into trainable bases, and interprets the meanings of bases by aligning them with diverse semantics. Each basis can be interpreted as a weighted combination of semantics (*e.g.,* face, female, and happy). By reformulating the saliency inference with a probabilistic method, the relationships between semantics and saliency can be quantified by integrating the model weights $W_{sal}$ (*i.e.,* the weight of each basis) and the semantic alignment $O$ (*i.e.*, the composition of semantics for each basis).

analyzes the features learned by saliency models by aligning activation maps with segmentation for a selection of objects (*e.g.,* body parts, food, and vehicles in [17]).

Our research contributes to the field of attention research by introducing a rigorous methodology for analyzing deep saliency models. In contrast to previous studies [18, 19, 42, 43], our study has three key distinctions: First, while previous analyses of deep saliency models were restricted to a predefined set of salient objects (*e.g.,* object segmentation used in [19]), our method automatically identifies both salient and non-salient semantics from a vocabulary of objects, parts, actions, and attributes. Second, different from previous qualitative analyses, our method quantifies the weights of these semantics in saliency prediction. It allows us to investigate the impacts of various factors (*e.g.,* the contributions of positive/negative semantics, the characteristics of datasets and model designs, and the process of fine-tuning) on saliency prediction, offering insights into the development of future deep saliency models. Third, our approach goes beyond analyzing the general visual saliency and demonstrates its strength in characterizing human visual attention under specific conditions such as the attention of people with autism spectrum disorder, the saliency of emotion-eliciting images, and time-course attention evolution.

## 3 Methodology

Human visual attention is influenced by a spectrum of visual cues, from low-level contrasts to high-level semantic attributes [5]. However, current deep learning-based saliency models [8, 9, 11] remain opaque in terms of the semantic attributes they have learned and how these attributes contribute to saliency prediction. To address this gap, we propose a method that decomposes neural network features into discriminative bases aligned with a wide range of salient or non-salient semantic attributes, and quantifies their weights in saliency prediction.

As illustrated in Figure 1, our method decomposes visual features by projecting them onto a collection of trainable bases, and uses the probabilistic distribution of bases to infer visual saliency. The overall idea is to identify both salient and non-salient semantics and quantify their impact on saliency prediction. To achieve this, we start with a deep saliency model and compare the features learned at the penultimate layer with different bases. This is done using a dot product between features $V \in \mathbb{R}^{M \times C}$ ($M$ and $C$ are the spatial resolution and dimension of features) and bases $B \in \mathbb{R}^{N \times C}$ ($N = 1000$ is the number of bases defined based on the number of units in the final layers of deep saliency models [9, 10]), which corresponds to their cosine similarity:

$$\alpha = \sigma(V \otimes B^T) \qquad (1)$$

where $\otimes$ denotes the dot product. $\sigma$ is the Sigmoid activation for normalization. $\alpha_{i,j} \in [0,1]$ represents the probability of $j^{th}$ basis $b_j$ detected at the $i^{th}$ region $P(b_j = 1|V_i)$. Inspired by [47] for decomposing model weights, we factorize the features as a weighted combination of matched bases:

$$V_i^f = \sum_{j=1}^{N} \alpha_{i,j} B_j \qquad (2)$$

where $V^f \in \mathbb{R}^{M \times C}$ are factorized features used to predict the saliency map $S = W^f V^f$ ($S \in \mathbb{R}^M$, $W^f$ are weights of the last layer).

Upon building the connections between visual features and discriminative bases, we then re-route the final saliency prediction by (1) freezing all model weights including the bases, and (2) adjusting the last layer for saliency inference based on the probabilistic distribution $\alpha$. We train a new layer (with weights $W^{sal}$) for predicting the saliency map:

$$S = \sum_{j=1}^{N} W_j^{sal} \alpha_{:,j} \qquad (3)$$

Intuitively, the method formulates the problem of saliency prediction as learning the linear correlation between the detected bases and visual saliency, which can be denoted as learning $P(S|b_1, b_2, ..., b_N)$. With the intrinsic interpretability of the design, *i.e.,* $\alpha$ as the probabilistic distribution of bases and $W^{sal}$ encoding the positive/negative importance of bases, we are able to investigate the weights of different bases to visual saliency.

The final step is to understand the semantic meanings of each basis. Unlike previous studies [18, 19, 42, 43] that focus on predefined salient objects, we take into account a comprehensive range of semantics without assumptions on their saliency. Specifically, our method leverages the factorization paradigm to measure the alignment between each basis and the semantics. Given an image and the regions of interest for different semantics (*e.g.,* bounding box annotations in Visual Genome [20]), we (1) compute the probabilistic map for each $j^{th}$ basis $\alpha_{:,j} \in \mathbb{R}^M$, and (2) measure its alignment $O_{j,p}$ with the regions of interest $R_p$ for each semantic $p$. Following [48], we binarize the probabilistic map with a threshold $t_j$ and measure the alignment with Intersection over Union (IoU):

$$O_{j,p} = \frac{|\mathbb{I}[\alpha_{:,j} > t_j] \cap R_p|}{|\mathbb{I}[\alpha_{:,j} > t_j] \cup R_p|} \qquad (4)$$

We use an adaptive threshold $t_j$ for each individual basis, which is defined to cover the top 20% of regions of probabilistic maps. Through iterating the measuring process for all images within the dataset, we are able to link bases learned from saliency prediction to a variety of visual semantics. We consider the top-5 semantics matched with each basis, and incorporate their average alignment scores $\hat{O}_{j,p}$ for determining the weight $I$ to saliency:

$$I_p = \frac{\sum_{j=1}^{N} W_j^{sal} \hat{O}_{j,p}}{Z} \qquad (5)$$

where $Z$ is the normalization factor that normalizes the contribution of the semantics to the range of [-1, 1] (for semantics with positive/negative contributions, Z denotes the maximal/minimum contribution among all semantics). This approach enables us to capture a broader range of contributing semantics, while avoiding an overemphasis on dominant salient/non-salient semantics (e.g., face and cloudness).

Overall, our method establishes the foundation for bridging implicit features learned by deep saliency models with interpretable semantics. It goes beyond existing studies that analyze models' predictive behaviors on a selection of object categories, as it considers a comprehensive range of semantics without assuming their relevance to saliency. It provides insights into how well the models capture both salient and non-salient semantics, and how they quantitatively contribute to saliency prediction.

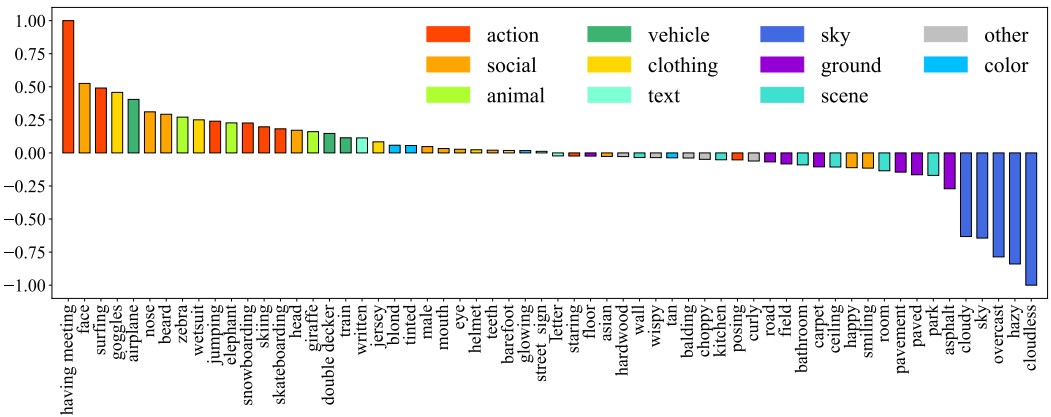

Figure 2: Important semantics learned by the deep saliency model and their weights. We visualize the top-60 semantics with significantly positive or negative weights.

## 4 Experiment

### 4.1 Implementation

**Semantic annotations.** To correlate implicit features with interpretable visual semantics, we leverage the Visual Genome [20] dataset. It has (1) multiple objects in the same scene to derive relative importance; and (2) a broad coverage of semantics in the naturalistic context, including objects, parts, attributes, and actions. We use the bounding box annotations of semantics (*i.e., $R$* in Equation 4) to measure the alignment between bases and semantics.

**Model configuration.** We experiment with three state-of-the-art saliency prediction models, including SALICON [9], DINet [11] and TranSalNet [10]. All models are optimized with a combination of saliency evaluation metrics (*i.e.,* Normalized Scanpath Saliency (NSS) [49], Correlation Coefficient (CC) [50], and KL-Divergence (KLD) [51]) as proposed in [8], and use ResNet-50 [13] as the backbone. Model training follows a two-step paradigm: (1) The model is optimized to factorize features with trainable bases, where the weighted combination of bases (*i.e., $V^f$* in Equation 2) is used for predicting the saliency map. Note that we do not use pretrained and fixed deep saliency models, but optimize the corresponding model architecture with the proposed factorization modules. (2) We freeze the model weights learned in the previous step and reroute the saliency inference to derive the saliency map from the probabilistic distribution $\alpha$ (see Equation 3) so that the interpretation is on features learned by the same saliency model. Only the last layer $W^{sal}$ is fine-tuned to learn the correlation between the distribution of bases and visual saliency.

### 4.2 How Do Deep Saliency Models Differentiate Salient and Non-Salient Semantics?

Deep saliency models are powerful tools for predicting visual saliency, and their ability to incorporate semantic information is crucial for closing the gap between computational modeling and human behaviors [17]. To gain insight into the semantics that deep saliency models learn and their contributions to saliency prediction, we apply our framework to the state-of-the-art DINet [11] trained on the SALICON [12] dataset (see Section 4.3 for results on other models and datasets), and explicitly measure the weights of diverse semantics during the inference process (*i.e., $I_p$* in Equation 5).

Figure 2 shows that the DINet model effectively captures a variety of semantics that are closely related to visual saliency. These include social cues such as faces, noses, and beards, actions like having a meeting, snowboarding, and jumping, clothing such as goggles, and salient object categories like animals, vehicles, and text. These findings resonate with previous research [18, 19, 43] that saliency models learn to recognize salient cues. More importantly, they showcase the versatility of our approach in automatically identifying key contributing factors of saliency models without any preconceived assumptions [43, 19], enabling attention analyses across a diverse array of scenarios (see Section 4.5).

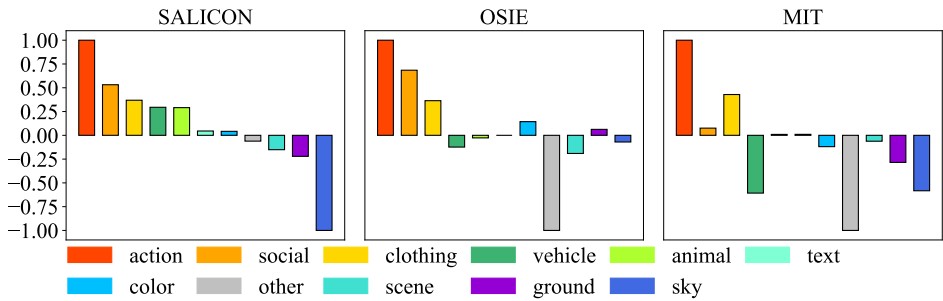

Figure 3: Semantic weights for DINet trained on different datasets.

Another unique advantage of our approach is to simultaneously derive semantics that both positively and negatively contribute to saliency, while previous studies commonly focus on the positive side. Our results reveal a clear separation between the semantics that contribute positively and negatively to saliency. Specifically, the model considers social and action cues to have a positive contribution to saliency, while semantics related to backgrounds such as sky (*e.g.,* hazy and overcast), ground (*e.g.,* pavement and carpet), and scene (*e.g.,* bathroom and park) have negative weights. The observation demonstrates that deep saliency models' success is not only due to the accurate detection of salient objects [18, 19, 43], but also strongly related to the ability to distinguish salient and non-salient semantics.

### 4.3 How Do Data and Model Designs Affect Saliency Prediction?

Training data and model designs play crucial roles in determining how well deep saliency models perform [18]. To understand these roles better, we conduct a comparative analysis of the effects of different training datasets and model architectures on saliency prediction.

Figure 3 compares the results of three DINet models trained on different datasets: SALICON [12], OSIE [17], and MIT [16]. It reveals that the shifts in semantics weights are tightly coupled with the characteristics of the training data. For instance, the model trained on OSIE pays more attention to social cues and non-salient semantics, because the OSIE dataset collects attention on semantic-rich stimuli with diverse social cues. Differently, the model trained on MIT assigns a less positive weight to social cues and a more negative weight to the vehicle category, which is likely due to the dataset's less emphasis on social semantics and the co-occurrence of vehicles and more salient objects. Therefore, differences in the semantic weights across models trained on different datasets reflect the variations in the semantics presented in the respective datasets.

Figure 4 compares the semantic weights of three different saliency models trained on the same SALICON [12] dataset: *i.e.,* DINet [11], SALICON [9], and TranSalNet [10]. While all models place the highest weights on actions and the lowest weights on ground, scene, and sky, the differences in model designs are reflected in their semantic weights. For instance, compared to the other models, TranSalNet considers clothing and text to be strongly salient but sky to be less non-salient. The results

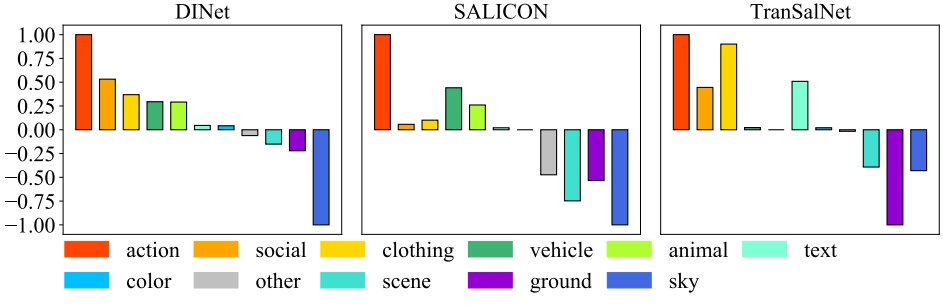

Figure 4: Semantic weights for different saliency models.

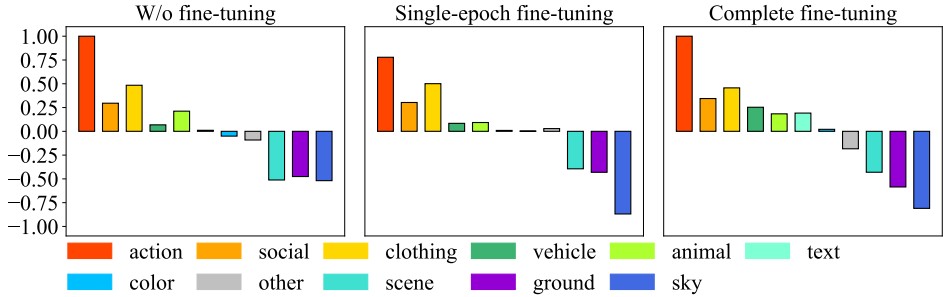

Figure 5: Evolution of semantic weights throughout the fine-tuning process.

shed light on the relationships between the designs and behaviors of saliency models, and show the usefulness of our framework in helping researchers tailor their models for specific applications.

Despite the differences in model behaviors across datasets and models, all models excel at automatically identifying salient semantics, *e.g.,* action and social cues, and differentiating foreground from background semantics. The observation is consistent with our findings in Section 4.2, and indicates that the ability to correlate semantics with saliency and quantify their positive/negative contributions is a systematic advantage of deep saliency models.

### 4.4 How Does Fine-tuning Saliency Models Affect Semantic Weights?

The process of fine-tuning is crucial for training deep saliency models, as highlighted in previous studies [18, 19]. To better understand the evolution of feature weights during fine-tuning, we conduct experiments on models trained in three scenarios of fine-tuning. These scenarios include models with fixed ImageNet [52] features (W/o fine-tuning), models that are fine-tuned for a single epoch (Single-epoch fine-tuning), and fine-tuned models with the best validation performance (Complete fine-tuning). We experiment with three different models, namely SALICON [9], DINet [11], and TranSalNet [10], and report the average results obtained from our experiments.

As shown on the left side of Figure 5, models with fixed ImageNet features already have the ability to identify salient semantics, such as action and social cues, highlighted with strong positive weights. This is consistent with previous findings [19], which validates the effectiveness of our approach.

When comparing the results of models with single-epoch fine-tuning (shown in the middle of Figure 5) to those without fine-tuning (shown on the left), we notice a significant shift in the negative weights. Specifically, while models with fixed ImageNet features identify scene-related semantics to contribute most negatively to saliency, after being fine-tuned for a single epoch, the models now focus on sky-related semantics and have a weaker emphasis on scene-related semantics. This can be attributed to the fact that ImageNet features are learned from iconic images, which are inherently insensitive to the sky background. However, since the sky is a strong indicator of low saliency values, it is necessary to learn this semantic to achieve accurate saliency prediction (note that the largest performance gain also occurs during the first epoch).

Finally, when examining the fully-tuned models (shown on the right side of Figure 5), we find that semantic weights continue to evolve during the later epochs of fine-tuning. Unlike the first epoch which imposes larger changes on a few negative semantics, the subsequent fine-tuning mostly plays a role in refining the weights of a broader range of semantics, enabling the models to become more sensitive to salient cues (*e.g.* action and text) and continuously adjusts the weights of negative semantics (*e.g.* ground and other).

These observations offer a comprehensive view of how deep saliency models progressively adapt ImageNet features through fine-tuning. They show that fine-tuning first concentrates on semantics with negative weights to saliency, which are not well captured in the pretrained features, and then gradually adjusts the relative weights of diverse semantics. Understanding the evolution of model behaviors during training can provide insights into optimizing the learning recipes for saliency models and enabling them to progressively encode the knowledge of diverse semantics.

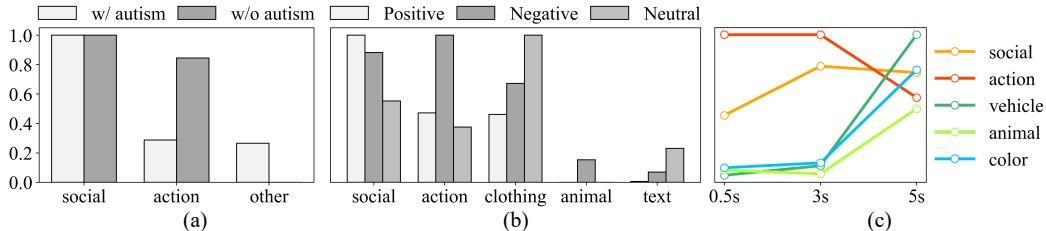

Figure 6: Semantic weights for characterizing the attention on three different settings. From left to right are results for the attention of people with and without autism, attention to stimuli eliciting different emotions, and the attention of different time periods. Figures share the same y-axis. Note that the results in the three figures are derived from different datasets selected based on the context of analyses.

## 4.5  Can Deep Saliency Models Capture Characteristics of Human Attention?

Human attention is influenced by several factors, such as the visual preferences of viewers, characteristics of visual stimuli, and temporal dynamics. We investigate the ability of DINet to capture the impact of these factors by training it on different attention data for each factor. We visualize results for the most discriminative semantics, and include the complete analysis in supplementary materials.

Firstly, we study the impact of visual preferences on attention with two subject groups, *i.e.,*, people with autism and those without, using a dataset with attention data of subjects from the two groups [53]. We train a DINet model on each set to compare their corresponding semantic weights. As shown in Figure 6a, both models assign significant weights to social cues, but the model for the autism group has a considerably weaker emphasis on action cues. This can be linked to the deficits in joint attention for people with autism [54, 55]. Additionally, the model for the autism group also assigns a strong positive weight to the "other" category. The specific objects highlighted in the category are related to gadgets, *e.g.,* digital, illuminated, and metal (see supplementary materials for details), which is consistent with the findings about the special interests of people with autism [53].

Next, we explore the impact of stimuli with diverse characteristics with images exhibiting positive, negative, and neutral sentiments, using the EMOd dataset [23]. Figure 6b shows that models trained on stimuli eliciting strong emotions (positive and negative) exhibit a higher emphasis on social and action cues (*e.g.,* face and nose, see our supplementary materials for details) than the one for neutral sentiment. Their behaviors align with previous findings that human attention generally prioritizes emotion-eliciting content over non-emotion-eliciting content (*e.g.,* clothing) [56, 57]. We also identify that models for positive/negative sentiments place more focus on human-related objects (*e.g.,* social cues and clothing) than objects less related to humans (*e.g.,* animals), which is a driving factor for characterizing the emotion prioritization effect on attention deployment.

Finally, we investigate the effects of temporal dynamics by conducting experiments on the CodeCharts1K [25] dataset that provides attention annotations collected at 0.5, 3, and 5 seconds. As depicted in Figure 6c, a shift of focus from dominantly salient cues (*e.g.,* action and social cues) to more diverse semantics (*e.g.,* vehicle, animal, and color) is depicted by the gradually increasing weights for the latter group. It is because viewers usually engage their attention to the most salient cues at the beginning of visual exposure (*i.e.,* within 3 seconds) before broadening their focus.

Overall, our findings suggest that deep saliency models can encode the fine-grained characteristics of diverse attention. They also validate the usefulness of our approach in revealing the discriminative patterns of attention and shed light on how visual attention is influenced by various factors.

## 4.6  What Is Missing to Close the Gap Between Saliency Models and Human Attention?

Previous studies [18, 43] have identified a collection of common mistakes for saliency models by probing the predicted saliency maps. In this paper, we aim to complement these studies by analyzing the failure patterns within the intermediate inference process using the proposed factorization framework.

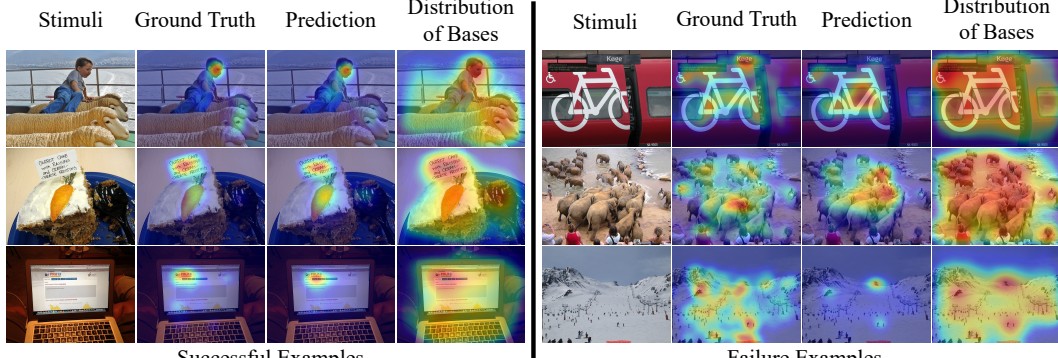

| Stimuli | Ground Truth | Prediction | Distribution of Bases | | Stimuli | Ground Truth | Prediction | Distribution of Bases |

Successful Examples          Failure Examples

Figure 7: Successful (left) and failure (right) examples of deep saliency models. We visualize the predictions from DINet, as those for the other two models are similar. For the distribution of bases, from red to blue, the probabilities of positive bases decrease while those for negative bases increase.

To conduct our analysis, we first select the common success and failure examples where three tested models (*i.e.,* SALICON [9], DINet [11], and TranSalNet [10]) consistently have high/low NSS scores [49]. Then we perform a qualitative analysis by visualizing the spatial probabilistic distribution ($\alpha$ in Equation 3) of the bases for semantics with positive (red) and negative (blue) weights (see our supplementary materials for details), which is used to derive the final saliency maps.

In the successful examples (see the left panel of Figure 7), we find that accurate saliency prediction correlates with the differentiation of diverse semantics. Specifically, stimuli for these examples typically have salient and non-salient regions belonging to different semantics. Therefore, with the ability to distinguish positive and negative semantics (*i.e.,* with discriminative distributions of the corresponding bases), models can readily determine their saliency distribution.

However, in the failure examples (see the right panel of Figure 7), models commonly struggle to determine the saliency within objects (*e.g.,* bicycle in the $1^{st}$ failure example) or among objects with similar semantics (*e.g.,* elephants in the $2^{nd}$ failure example). Investigation of the probabilistic distribution of bases shows that models typically have a uniform-like distribution of bases on object parts or among objects of the same category, thus inherently incapable of determining the importance of the bases to construct an accurate saliency map. We also note that existing models have difficulty with scenes without salient objects, which is illustrated in the $3^{rd}$ failure examples with a relatively empty scene.

As shown in Figure 7) (second column of the right panel), the ground truth human attention of the failure patterns is scattered. We further look into the inter-subject variability of these cases and found that their inter-subject variability is high, suggesting that human viewers may not agree on where to look, and therefore the ground truth maps are less valid. In this case, one assumption about saliency modeling (i.e., certain commonality about human attention patterns) may not be true, and the validity of using the ground truth human map for training and evaluation (i.e., the standard leave-one-subject-out approach), and the expected behavior of targeted models are interesting and open questions.

Overall, while high-level semantics learned in existing deep saliency models are powerful, we hypothesize that leveraging more structured representations to encode the contextual relationships between semantics, and integrating mid- and low-level cues will be helpful for accommodating challenging scenarios in saliency prediction.

### 4.7 Quantitative Evaluation of the Interpretable Model

The fundamental objective of our study is to develop a principled framework for understanding the underlying mechanism behind deep saliency models without altering their inherent behaviors. For this, we only introduce minimal architectural modifications, limited to the last two layers of the saliency models, thereby ensuring that their performance aligns seamlessly with the original models across all datasets. To complement our aforementioned analyses and further substantiate the efficacy

|  | OSIE [17] | | MIT [16] | | SALICON [12] | |
|---|---|---|---|---|---|---|
|  | NSS | CC | NSS | CC | NSS | CC |
| SALICON [9] | 2.75 | 0.63 | 2.56 | 0.70 | 1.89 | 0.86 |
| SAM [8] | 2.70 | 0.65 | 2.47 | 0.69 | 1.84 | 0.86 |
| DINet [11] | 2.88 | 0.63 | 2.54 | 0.70 | 1.92 | 0.87 |
| Ours | 2.91 | 0.64 | 2.53 | 0.70 | 1.89 | 0.86 |

Table 1: Comparative results of saliency prediction on three popular datasets. Models are evaluated with two common metrics, including Normalized Scanpath Saliency (NSS) [49] and Correlation Coefficient (CC) [50].

of our methodology, we quantitatively evaluate the saliency prediction performance of our method (using a DINet backbone trained the SALICON training split as an example) on three commonly used saliency datasets, including OSIE [17], MIT [16], and SALICON [12]. Comparative results reported in Table 1 demonstrate the competitive nature of our approach with respect to state-of-the-art methods, and validate its effectiveness in achieving the balance between interpretability and model performance.

## 5    Conclusion, Limitations, and Broader Impacts

As deep saliency models excel in performance, it is important to understand the factors contributing to their successes. Our study introduces a novel analytic framework to explore how deep saliency models prioritize visual cues to predict saliency, which is crucial for interpreting model behavior and gaining insights into visual elements most influential in performance. We discover that the models' success can be attributed to their accurate feature detection and their ability to differentiate semantics with positive and negative weights. These semantic weights are influenced by various factors, such as the training data and model designs. Furthermore, fine-tuning the model is advantageous, particularly in allocating suitable weights to non-salient semantics for optimal performance. Our framework also serves as a valuable tool for characterizing human visual attention in diverse scenarios. Additionally, our study identifies common failure patterns in saliency models by examining inference processes, discusses challenges from both human and model attention, and suggests modeling with holistic incorporation of structures and lower-level information.

Despite advancing attention research from different perspectives, our work still has room for improvement. Specifically, the current study focuses on visual attention deployment in real-world scenarios and employs the commonly used natural scenes as visual stimuli. We are aware that certain applications (*e.g.,* graphical design and software development) may also involve artificial stimuli, such as advertisements, diagrams, and webpages, and extension of the proposed framework to broader domains can be straightforward and interesting.

We anticipate several positive impacts stemming from this research. Firstly, by advancing the understanding of deep saliency models, valuable insights can be applied to optimize interfaces for human-computer interactions. This, in turn, will enhance the efficiency and reliability of the next generation of computer-aided systems, leading to improved user experiences and increased productivity. Secondly, the accurate capture of visual importance can have significant implications for individuals with visual impairments. Saliency models can effectively assist visually impaired individuals in navigating and interacting with both people and environments. This could empower them to engage more fully in various aspects of daily life, fostering independence and inclusivity. In sum, this comprehensive exploration of computational saliency modeling holds great potential for broader societal benefits.

## Acknowledgements

This work is supported by NSF Grants 2143197 and 2227450.

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
