# What Do Deep Saliency Models Learn about Visual Attention? (Supplementary Materials)

**Shi Chen**     **Ming Jiang**     **Qi Zhao**
Department of Computer Science and Engineering,
University of Minnesota
{chen4595, mjiang, qzhao}@umn.edu

The supplementary materials provide additional results to complement our analyses in the main paper, and elaborate on the implementation details of our visualization method.

## 1 Supplementary Results

In the main paper, we visualize the weights of different semantic categories (*e.g.,* action, social, and scene) for saliency prediction in various scenarios. Here we provide complementary results on detailed semantics, which are used to derive the results shown in the main paper (see the listed sections below). In particular,

- Figure 1 shows the weights of detailed semantics for DINet [1] trained on different datasets (Section 4.2 of the main paper).
- Figure 2 shows the weights of detailed semantics for models with three different architectural designs trained on the SALICON [2] dataset (Section 4.3 of the main paper).
- Figure 3 shows the weights of detailed semantics for models trained on the attention of people with and without autism (Section 4.4 of the main paper).
- Figure 4 shows the weights of detailed semantics for models trained on stimuli eliciting positive, negative, and neutral emotions (Section 4.4 of the main paper).
- Figure 5 shows the weights of detailed semantics for models trained on attention collected at different time periods (Section 4.4 of the main paper).
- Figure 6 summarizes the results in Figure 3, 4, and 5, by visualizing their corresponding weights of semantic categories.

## 2 Supplementary Details for Visualizing the Distribution of Bases

To identify the underlying rationales for the successes and failures of saliency models, in the main paper, we perform a qualitative analysis by visualizing the probabilistic distributions of trainable bases. In this section, we provide the details of the visualization approach.

Specifically, we first identify the top-10% of bases that contribute the most positively or negatively to the saliency map. The contributions are based on the model parameters in the final layer (*i.e.,* $W_{sal}$ in Equation 3 of the main paper). After identifying these most impactful bases, we compute their corresponding probabilistic distributions (*i.e.,* $\alpha$ in Equation 1 of the main paper), and spatially accumulate the distribution for bases at different positions. The distributions for positive and negative bases are weighted by multiplying 1 or -1, respectively, to highlight their polarities with different colors. Finally, we normalize the spatial distribution to range $[-1, 1]$ and overlay it with the original image for visualization.

37th Conference on Neural Information Processing Systems (NeurIPS 2023).

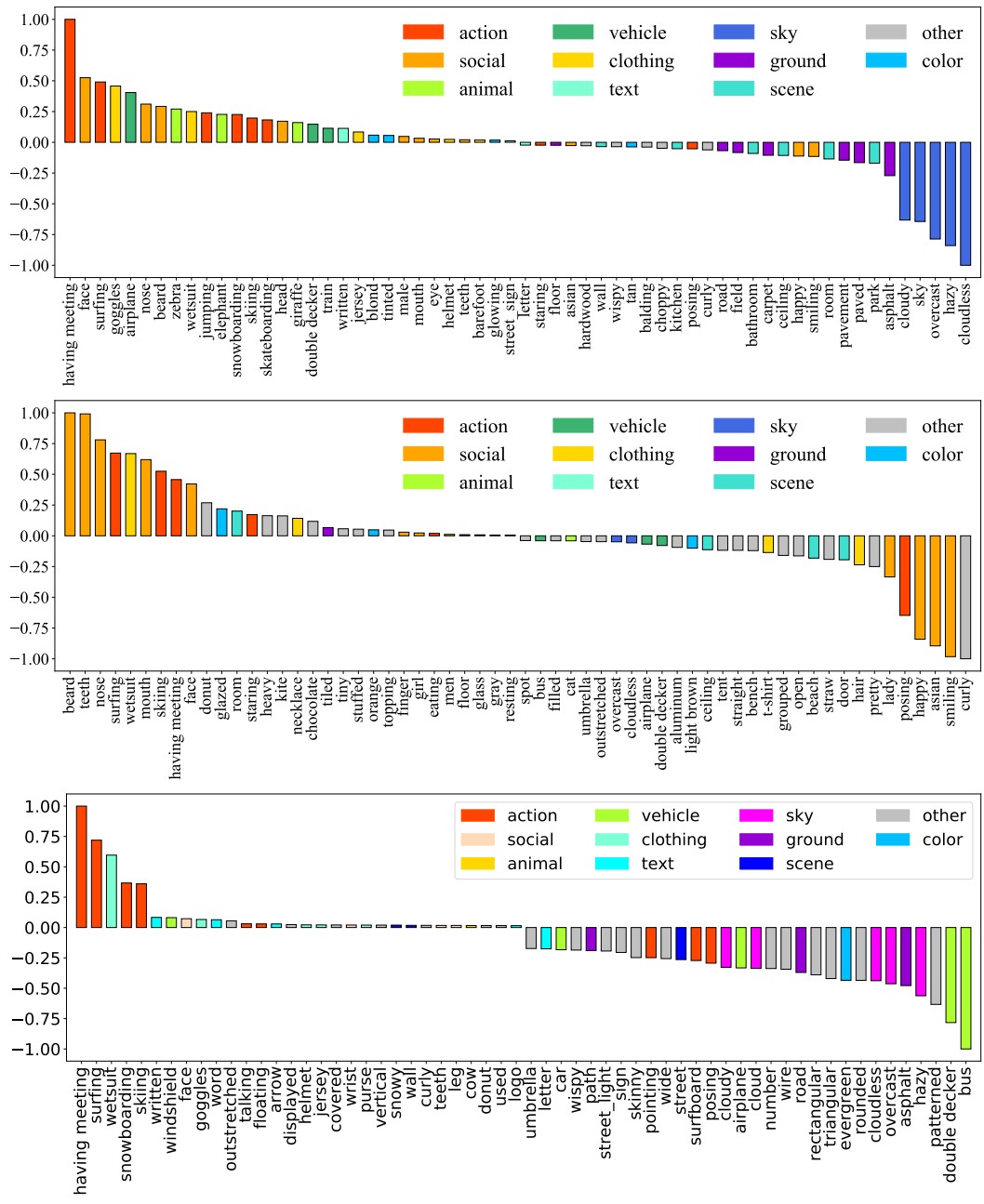

Figure 1: Weights of detailed semantics for DINet trained on SALICON [2] (top), OSIE [3] (middle), and MIT [4] datasets (bottom).

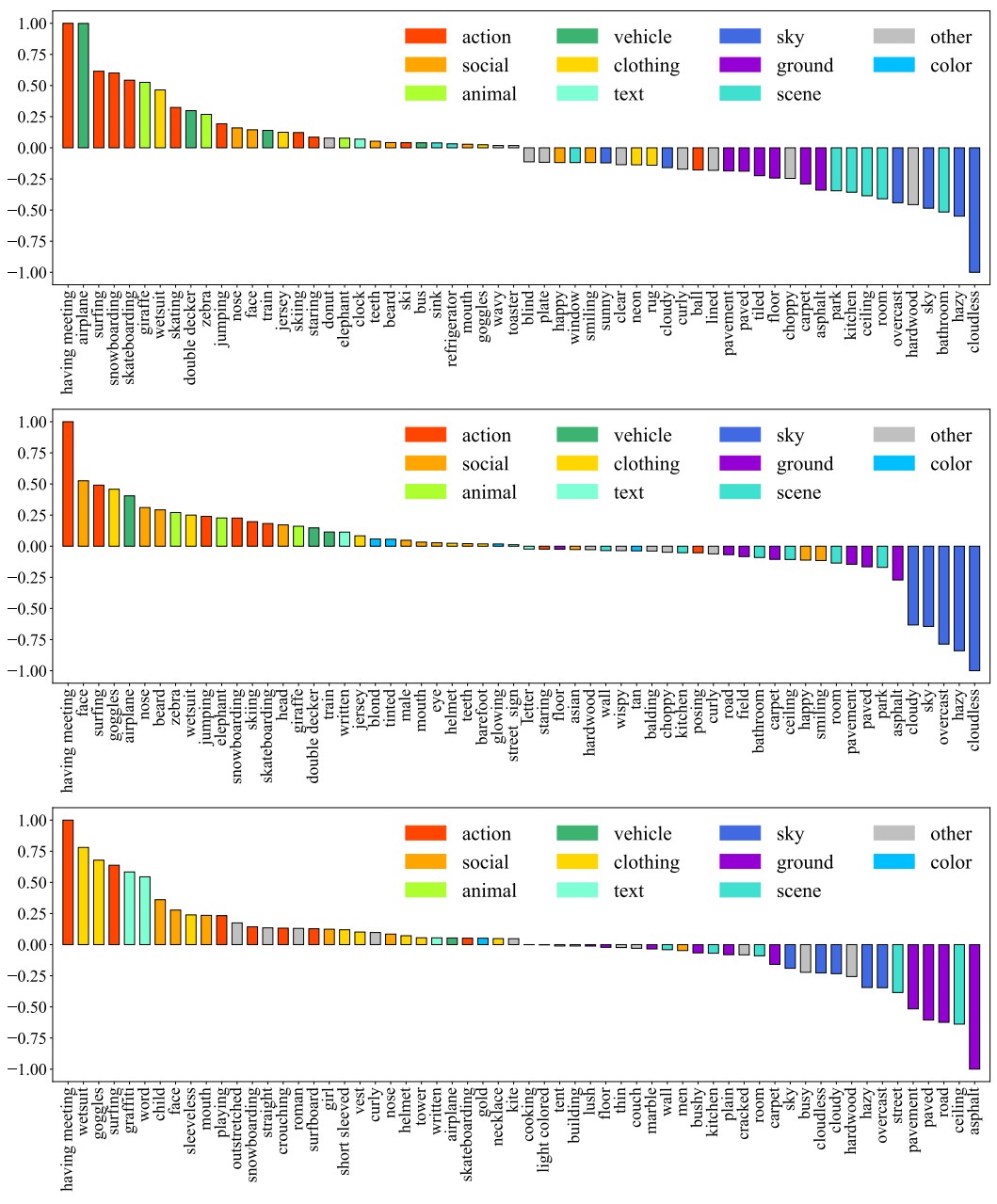

Figure 2: Weights of detailed semantics for SALICON [5] (top), DINet [1] (middle), and TranSalNet [6] (bottom). All models are trained on the SALICON [2] dataset.

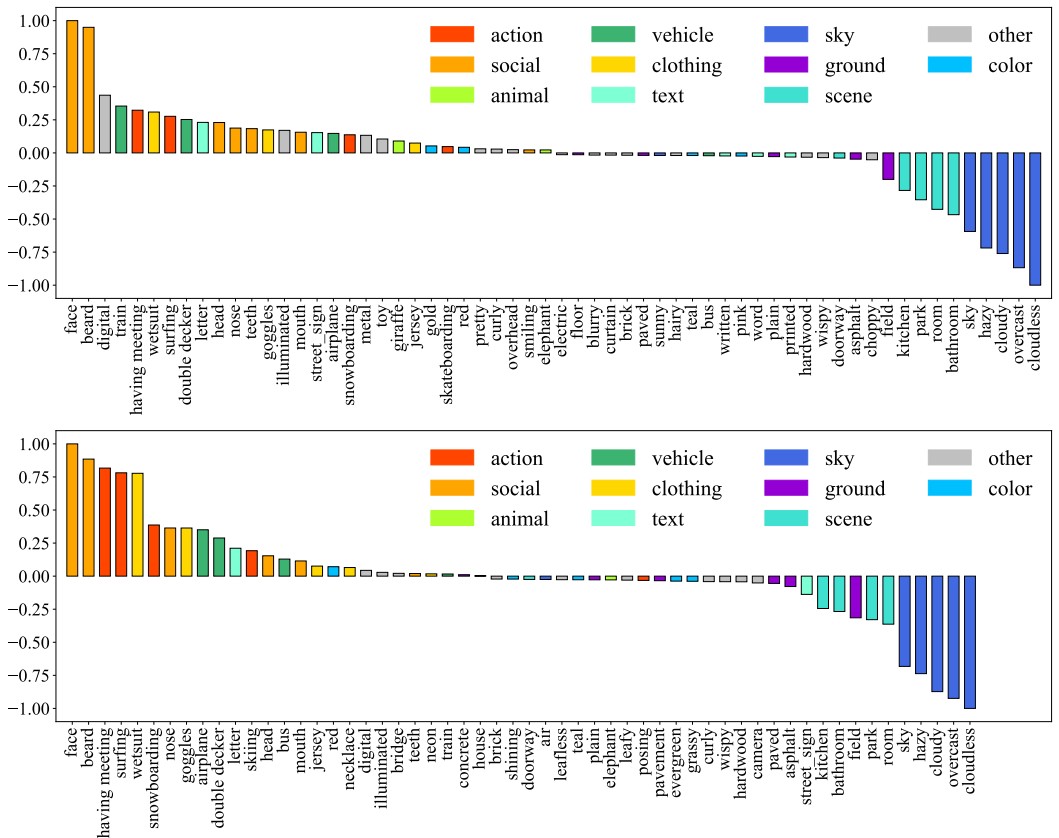

Figure 3: Weights of detailed semantics for models trained on attention for people with (top) and without (bottom) autism. Experiments are conducted on the eye-tracking dataset proposed in [7].

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

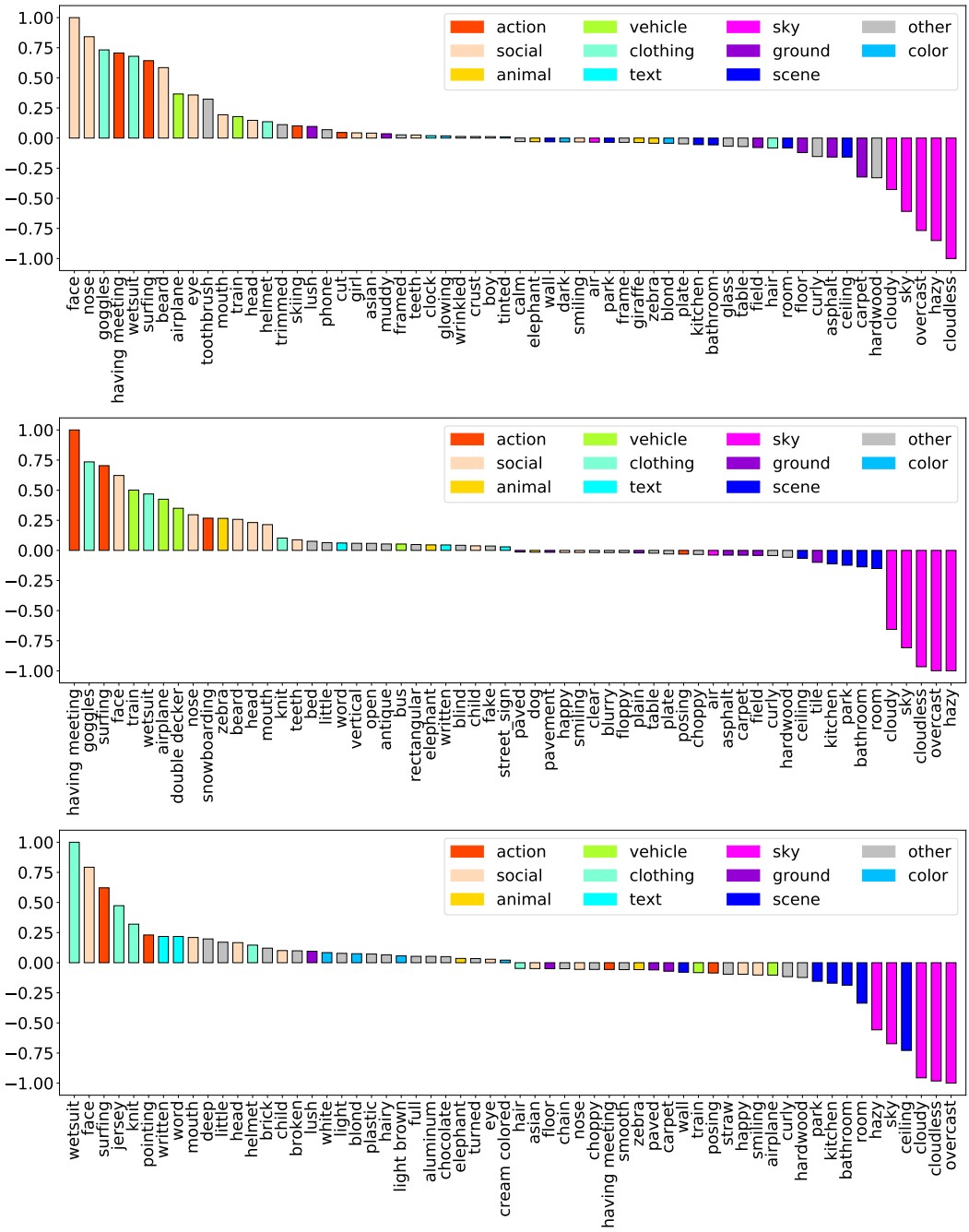

Figure 4: Weights of detailed semantics for models trained on stimuli eliciting positive (top), negative (middle), and neutral (bottom) emotions. Experiments are conducted on the EMoD [8] saliency dataset.

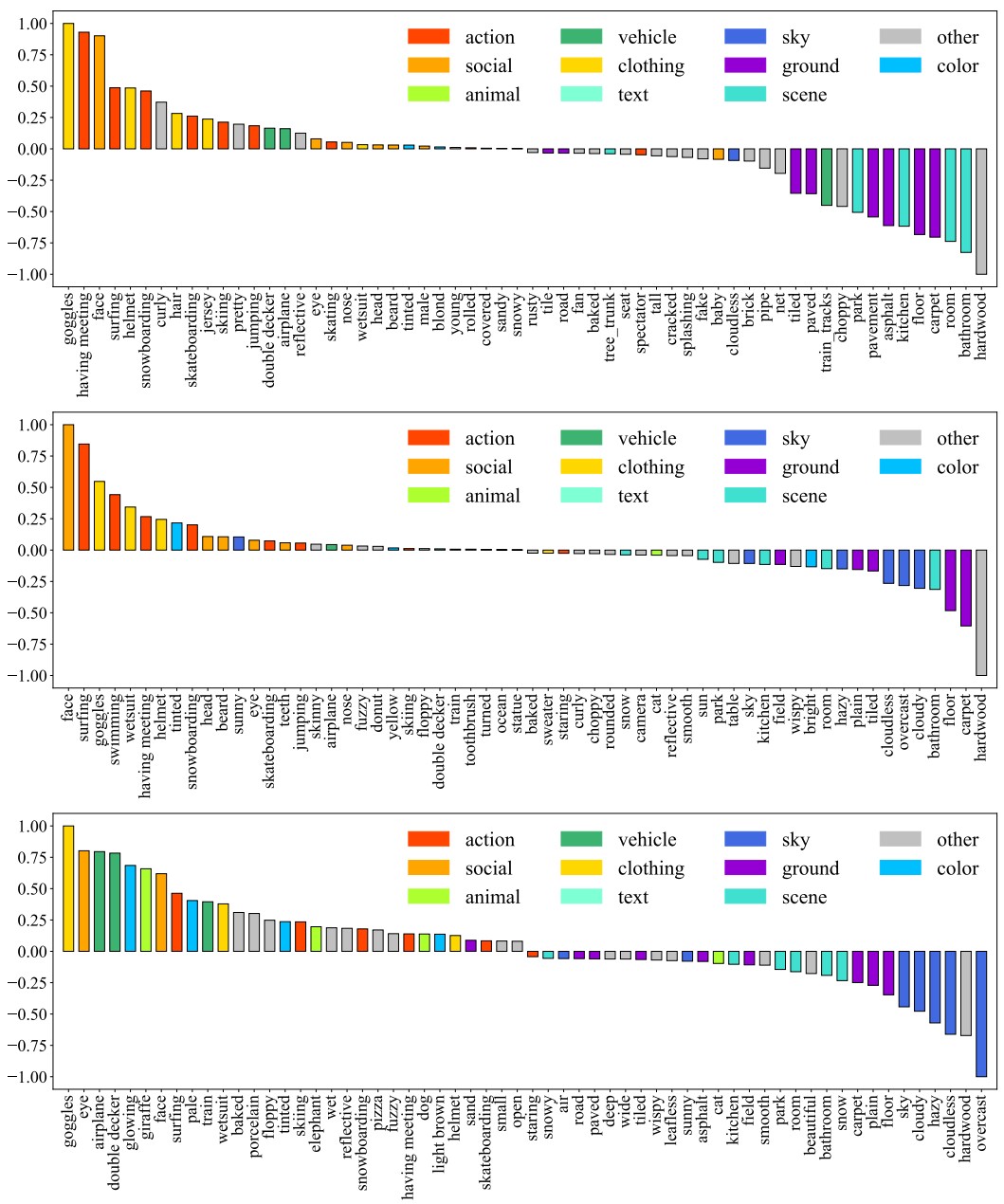

Figure 5: Weights of detailed semantics for models trained on attention collected at 0.5 (top), 3 (middle), and 5 (bottom) seconds. The data is adopted from the CodeCharts1K [9] dataset.

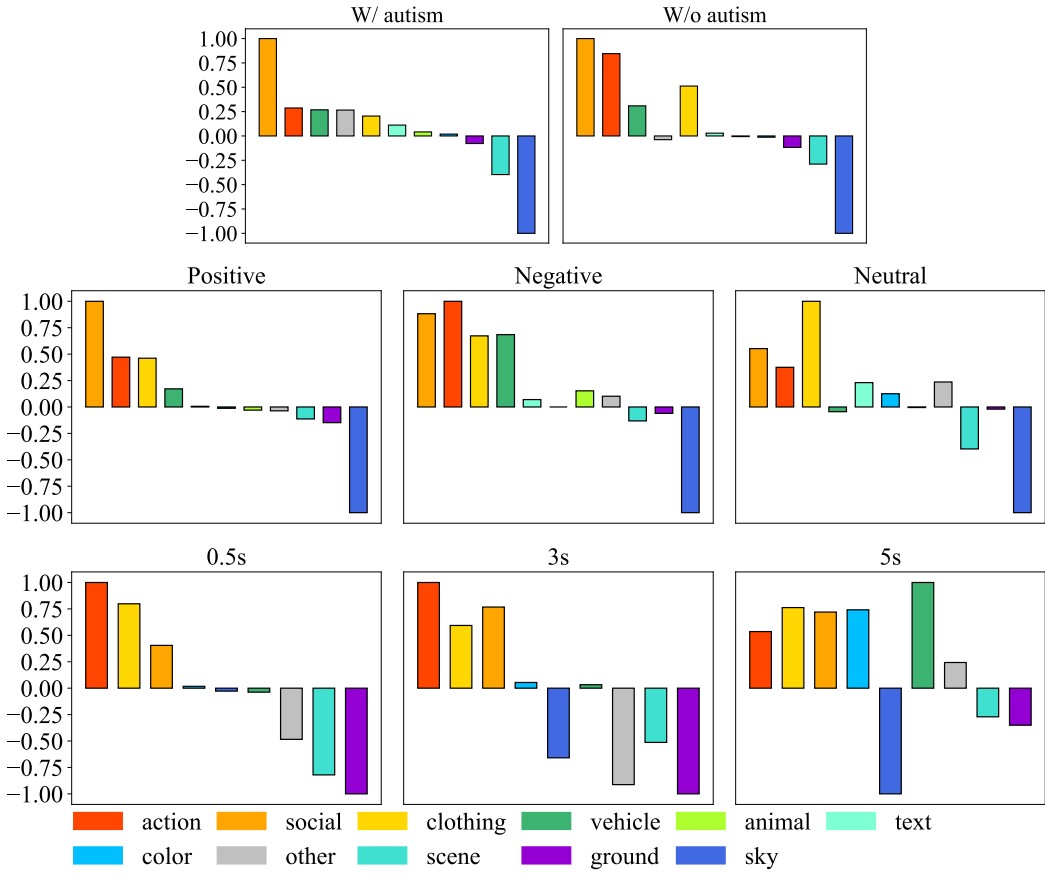

Figure 6: Weights of different semantic categories for attention in three different scenarios. Specifically, from top to bottom are the attention for people with or without autism, attention for stimuli eliciting positive, negative, and neutral emotions, and attention collected at different time periods.