# OpenReview forum: "What Do Deep Saliency Models Learn about Visual Attention?"
_NeurIPS.cc/2023/Conference — NeurIPS 2023 poster_

### Official Review · Reviewer_UKcT · 2023-07-03

**Soundness:** 2 fair
**Presentation:** 3 good
**Contribution:** 2 fair
**Rating:** 4
**Confidence:** 4

**Summary:**

This paper proposes a new framework, which can decompose the learned features of a saliency model into trainable bases (using [42]), those bases are combined to formulate the final saliency map, and the weight of the combination indicates the contribution of each basis. The semantic meaning of each basis can be explored by matching the bases to a probe dataset (Visual Genome is used in this draft), Thus, saliency can be studied based on visual concepts, .e.g, different types of visual relationship, or objects, exploring what the key factors are behind the saliency region.

**Strengths:**

This draft is well written, the organization is clear, and the method is explained in details. The proposed method attempts to solve a fundamental and important question, what features are important to saliency.



**Weaknesses:**

I like this study attempts to solve a very interesting question, however, some of the designs may need more discussion. This study is based on the existing basis de-composition[42] technique. The main contribution is to link those bases to other visual concepts by computing thresholded IOU. This design heavily relies on the probing dataset, which makes the hyper-parameter threshold a little tricky. What if other datasets are used, will the bases also match other concepts, how one can validate the matched concept (objects or events) is valid. Some studies attempt to explore the learned feature based on known probing concepts, e.g., designing a dataset specifically for texture and shape concepts(Geirhos, Robert, et al. 2018). Exploring visual concepts (bases) by matching to other datasets is not very convincing. Should one exhaustively match to all concepts to find the best match (I highly doubt if this is feasible or how to define the universe), or setting a threshold and matching to a “local”’ solution(a specific dataset)? Thus, matching bases to a specific dataset is not very solid to show what the bases are.

More generally, assuming we can find the “ground-truth” (the true meaning of those bases), I believe saliency is way more complicated than simply based on visual concepts. From Figure 2, we can see that street sign is less important than faces. We can imagine the street sign could be the most salient region if the picture only contains a street sign in the middle. Given two images, one has a face in the middle and one has a street sign in the middle, they could be equally salient. Thus, one explanation for figure 2 could also be it shows the bias of the dataset which used to train the model, more faces appeared in that dataset. There also could be biases in the probe dataset, Visual Genome in this study. One concept, say “having meeting”, is more salient could be the visual features happen to match one of the bases. The real most salient basis does not exist in that dataset.

Another problem in this study is that saliency may not depend on the semantic. Imagine we have ten faces in a picture, the middle could be the most salient, while the others are not. Thus, study which visual basis is the most salient is not convincing. This may also happen to the Visual Genome dataset. The region for the visual relation is the union of the head and the tail, which is larger than the two objects. The bigger union region may better correlate with the well-known centre bias regardless of what the actual content. This is why “action” is the most salient in all of the datasets in Figure 3.

Overall, applying basis decomposition [42] is not new, the finding of the semantic importance given saliency may simply showing dataset biases.
Geirhos, Robert, et al. "ImageNet-trained CNNs are biased towards texture; increasing shape bias improves accuracy and robustness." arXiv preprint arXiv:1811.12231 (2018).


**Questions:**

See above

**Limitations:**

Exploring unknown bases to a dataset is questionable, which requires further study. The finding of those bases may not explain saliency, but the biases of the training and probing datasets.

---

> ### Author Rebuttal · Authors · 2023-08-10
>
> **1. Q**: What if other datasets are used, will the bases also match other concepts? Should one exhaustively match all concepts to find the best match?
>
> **R**: We appreciate the thoughtful questions raised about the potential limitations of matching bases to a single dataset. We agree that the matching of bases and concepts may depend on specific datasets or thresholds used, and it is challenging to define or exhaust all concepts in the universe. Our study follows the common practice of leveraging big datasets with key concepts at scale [42, 43]. In particular, we use Visual Genome, the biggest and most diverse naturalistic dataset with a sufficiently large number of images (100,000 images) and fine-grained annotations, providing a rich and diverse set of objects, scenes, attributes, relationships, and contexts. Our approach automatically captures salient and non-salient semantics (Figure 2) that agree with previous studies [19, 40, 41], as well as discriminative attention patterns [49-52] in various contexts (e.g., for different participant groups, stimuli, and time durations in Figure 6). These findings validate the effectiveness of our method in drawing key insights into saliency.
>
> **2. Q**: I believe saliency is way more complicated than simply based on visual concepts. Given two images, one has a face in the middle and one has a street sign in the middle, they could be equally salient.
>
> **R**: Saliency suggests relative importance and thus the respective capability in attracting attention when multiple objects or semantics co-occur in a scene.  We agree with the reviewer that it is dependent on other factors such as the context of an image and the locations of the objects. We also agree that with a face in the middle and a street sign in the middle, both are salient in their individual contexts. It is in fact difficult to understand the relative importance of objects or semantics with iconic images with one or few dominant objects in the center, and this is indeed the reason that we leverage datasets with images that (1) have multiple objects co-occurring in images, and (2) include diverse objects in naturalistic context; so the confounder effects such as center bias are neutralized to some degree, and statistical conclusions of their importance can be derived. Therefore, instead of looking at saliency in an individual image, our framework derives conclusions from a more global perspective from a large set of naturalistic images with various objects and contexts.
>
> **3. Q**: One explanation for Figure 2 could also be it shows the bias of the dataset which used to train the model, more faces appeared in that dataset.
>
> **R**: Our method uses the average IoU score across diverse images annotated with a specific concept to measure its importance in Equation 5, which accounts for the different frequencies of concepts. Therefore, higher frequencies of a concept do not necessarily result in higher importance. For instance, the "mouth" has an equivalent frequency to "face" but is assigned much lower importance, the “floor” has fifteen times more frequency than “cloudless” but their importance shows the opposite.
>
>
> **4. Q**: The finding of those bases may not explain saliency, but the biases of the training and probing datasets.
>
> **R**: We acknowledge that both the training dataset and the probing dataset may introduce biases in the saliency analysis. However, the large-scale Visual Genome dataset mitigates the risk of biases from limited data samples present in smaller-scale studies. This approach helps in understanding the general strategies employed by deep saliency models for capturing attention across diverse visual contexts. As highlighted in Section 4.5, our method results in meaningful results that are consistent with human vision studies [49-52], validating its effectiveness in explaining saliency instead of biases.
>
> **5. Q**: The region for the visual relation is the union of the head and the tail, which is larger than the two objects. The bigger union region may better correlate with the well-known center bias regardless of the actual content.
>
> **R**: Our approach takes advantage of two key strategies to counter center biases and the size of semantics, and thus ensure a faithful measurement of the alignment between the probabilistic distribution of a semantic and the segmentation of different semantics.
>
> First, size-insensitive measurement. We follow the general methodology of [43] and measure the alignment with the IoU score and adaptive thresholding (i.e., thresholds that capture the top 20% quantile level of distributions for different bases). This method ensures that the alignment score will be high only if the probabilistic distribution of the basis aligns with the majority of the segmentation mask, regardless of the average size of the semantics. For instance, on average, a "train" is over seven times larger than a "face", but our approach correctly recognizes the significantly higher importance of "face" in Figure 2.
>
> Second, the choice of dataset.  Unlike many image datasets (e.g., ImageNet and MIT) focusing on iconic images where few objects are centered, Visual Genome used in our study is designed to encompass a rich set of semantic concepts in the same scene. Due to a more balanced distribution of visual features, the most salient object is not always at the center of the scene. Therefore, this design alleviates center biases.

---

> > ### Comment · Reviewer_UKcT · 2023-08-14
> > **feedback**
> >
> > Thanks the authors for the further explanation, however, not all of the questions are answered in detail.
> > 1. " Visual Genome, the biggest and most diverse naturalistic dataset with a sufficiently large number of images". I looked into the VG dataset, which is also very skewed, 90% relations belong to the categories of possessive(has part of) and geometric(above behind). More importantly, "people" and "vehicles" are also dominant in the objects. Thus, it is difficult to say the VG dataset is "large" (or "fair"?) enough to test saliency. Moreover, it is unknown to me how to define a dataset is large enough for this purpose. This also explains the use of the adaptive threshold. The true unknown concept does not exist, so a lower threshold is used to select the second or third best match.
> > 2.b "include diverse objects in naturalistic context; so the confounder effects such as center bias are neutralized to some degree,". I looked in to the VG data, I cannot see how the diverse objects can solve the centre bias effect. In contrast, the definition in visual scene graph could worsen this problem. The visual feature of relation, "having meeting", the union region of two objects correlates more with centre bias, which explains why the relationship is more salient, instead of reducing the center bias effect.
> > 3. Thanks for the explanation, this could partially answer the frequency in the probing dataset. Could you please add a little more (maybe one sentence about this frequency normalization Z), is this new in this study? I might miss it in [43].
> > 4. Again, it is still unclear how to define VG is large enough for this purpose, and the relation (union region) definitely correlates more with the saliency region due to the center bias.
> > 5. a) This size-intensive measure Eq.4 cannot handle the center bias problem. Imagining that most saliency appears more in the center of an image, the union region also has this tendency comparing with the head and tail objects. Thus, Eq4 could easily capture relation (union region) than single objects regardless of what threshold is applied.
> > b). VG dataset is not balanced though it has a lot of categories. More importantly, the VG data also has centre bias due to the human bias, and the annotation (relation) is created by human where relation around RoIs(center) is labeled.
> > Thus, both measures are not used for the position bias.

---

> > > ### Author Response · Authors · 2023-08-15
> > > **Response to feedback (part 1)**
> > >
> > > We appreciate the reviewer for comments and exchange of ideas. Center bias, imbalance categories, and dataset selection are indeed important considerations for not only our research but the whole community. Please find below our elaborations on these issues:
> > >
> > > **1. Q**: How do diverse objects solve the center bias effect? 90% relations belong to the categories of possessive(has part of) and geometric(above behind). Does the relation (union region) definitely correlate more with the saliency region due to the center bias?
> > >
> > > **A**: We appreciate the discussions about center bias. We agree with the reviewer that center bias naturally exists in the datasets due to human bias (e.g., rules of composition in photography). While the majority of studies or methods live with such bias, we are aware of it and identified ways to counter unwanted bias, from both dataset selection and method design perspectives as elaborated below:
> > >
> > > **Dataset selection**: compared with conventional iconic images with one or few dominant objects in the center, some later datasets highlight multiple salient objects in a naturalistic context, which reduces center bias. For example, the OSIE dataset [17] was proposed to counter center bias by carefully selecting stimuli with a large number of salient off-center objects, demonstrating a much smaller center bias with popular datasets like the MIT saliency dataset [16]. To demonstrate the relatively small center bias on the Visual Genome dataset, we report the key indicators evaluating center bias in comparison with OSIE.
> > >
> > > | | Number of Objects Per Images | Object Distance to Center|
> > > |----------|----------|----------|
> > > | OSIE | $ 7.93 \pm 3.95$ | $0.3 \pm 0.13$ |
> > > | Visual Genome| $16.43 \pm 8.21$ | $0.3 \pm 0.14$ |
> > >
> > > **Method development**: We would like to clarify several efforts to counter center bias in our method: (1) Our analyses are based on objects and their attributes (including action-related attributes) obtained from Visual Genome. We do not consider object relations in our pool of semantics, and thus our analytic framework is not affected by the effects of object unions. (2) Our method leverages the IoU metric to measure the alignment between probabilistic distributions of a basis and the segmentation of different semantics, thus interpreting the semantic meaning of the basis. The metric is independent of the position of semantics, and counters the effect of size by normalizing with the union between probabilistic distributions and segmentation of semantics. As a result, it is insensitive to the larger semantics appearing in the center. Our further analyses confirm that the semantic importance has weak correlations with distance to the image center (**Pearson’s r=-0.05**) or object size (**Pearson’s r=-0.11**), which underscores that large objects or union regions near the center do not necessarily imply higher importance.
> > >
> > > **2. Q**: VG dataset is very skewed (imbalanced), though it has a lot of categories.
> > >
> > > **A**: We agree with the observation regarding the skewed distribution. Like center bias, imbalance categories exist in most naturalistic datasets. We would like to add several observations:
> > >
> > > (1) On the positive side, the skewed distribution reflects the frequencies of objects or semantics occurring in real life, offering opportunities to develop insights and models for the wild.
> > >
> > > (2) Recent studies (e.g., [42, 43]) have commonly leveraged such large skewed naturalistic datasets as probing datasets, and provided valuable insights into the behaviors of deep networks.
> > >
> > > (3) We follow [43] and incorporate adaptive thresholding to maintain the stability of relative ordering among semantics, regardless of their size or frequency. Furthermore, we incorporate an averaging process that divides the alignment scores $O$ in Eq.4 by the number of images, effectively countering the imbalanced distribution. Such a paradigm enables us to take into account the naturalistic contexts without overfitting to data biases.
> > >
> > > **3. Q**: Could you please add a little more on the normalization Z, is this new in this study? I might miss it in [43].
> > >
> > > **A**: The reviewer is right that Z is new in this study. [43] interprets a deep layer for image classification and does not consider normalization. Differently, our study aims to unveil the nuanced relationship between semantics and visual saliency, and it for the first time quantifies the contributions of semantics with deep saliency models. Besides considering Z that normalizes the contribution of the semantics to the range of [-1, 1] (for semantics with positive/negative contributions, Z denotes the maximal/minimum contribution among all semantics), the numerator part of Eq. 5 is also new, which quantifies the contribution of semantics to saliency by incorporating the semantic meanings of bases and their corresponding weights learned in deep saliency models. We will incorporate the details in the revision.

---

> > > > ### Author Response · Authors · 2023-08-15
> > > > **Response to feedback (part 2)**
> > > >
> > > > **4. Q**: It is difficult to say the VG dataset is “large” enough to test saliency. It is unclear how to define a dataset large enough for the analysis.
> > > >
> > > > **A**: It is indeed an interesting and open question as to how to decide whether a dataset is large enough, which would depend on a number of factors covering dataset properties and tasks. Saliency research does not demand huge datasets and it has made significant progress with datasets at ~1000 image scale (e.g., MIT [16] and OSIE [17]) and then ~10,000 image scale (e.g., SALICON [12]) over the last decade. It is important, however, to our research goal here that the dataset has (1) multiple objects in the same scene to derive relative importance; and (2) a broad coverage of semantics in the naturalistic context so our conclusions can provide meaningful insights aligned with humans or lead to models that work in the real world context. While one can always hope for a bigger and better dataset, Visual Genome, with ~100,000 naturalistic images, more than 700 semantics, and on average 16 objects per image, is a reasonable choice for this study.

---

> > > > > ### Author Response · Authors · 2023-08-16
> > > > > **Update of previous response**
> > > > >
> > > > > We realized that there may be a misunderstanding with "relation" that was mentioned in a few places in the comments. We addressed it and also updated our responses to other questions. Thank you very much for your insightful discussions!

---

### Official Review · Reviewer_srbe · 2023-07-05

**Soundness:** 3 good
**Presentation:** 3 good
**Contribution:** 3 good
**Rating:** 6
**Confidence:** 4

**Summary:**

This paper examines the problem of predicting visual saliency in images. Unlike many other works, it focuses on determining what leads to the predictions made, including underlying features that are learned, and formulating the prediction as a combination of bases. Through this, one is able to garner an understanding of how the models make their decisions and attach this to semantic information or other concepts.

**Strengths:**

Strengths of this paper are as follows:
1. It approaches saliency from a different point of view; rather than competing to marginally beat an ROC score, it takes a step towards truly understanding the success of deep learning models in this domain
2. By explicitly modelling features correlated with attention, and those inversely correlated with attention, one can model both what is salient and what is not in a way that allows these concepts to be quantified
3. Because the model constructs it's prediction from a set of bases, this effectively allows human attention to be modelled subject to a change to these bases, as demonstrated in an example involving autism.

**Weaknesses:**

Ultimately, it would be nice to see some quantitative results on how well the model performs compared to some of the state-of-the-art models, although I understand this is not the point of the paper. Nevertheless, a detailed set of results along this dimension could add value but I wouldn't insist on it.

**Questions:**

1. If possible, can you comment on how this formalism for saliency prediction extends to other domains?

2. In figure 5, it seems that a single epoch of fine tuning is closer to careful fine tuning than the original weights. Is this from using a relatively large learning rate?

3. Related to the above, how does this vary with changes to the learning rate?


**Limitations:**

The authors have addressed limitations of their approach in Section 5 of the paper.

---

> ### Author Rebuttal · Authors · 2023-08-10
>
> **1. Q**: It would be nice to see some quantitative results on how well the model performs.
>
> **R**: We thank the reviewer for the suggestion and include the results in the global comment, which shows that our model achieves state-of-the-art performance.
>
> **2. Q**: Can you comment on how this formalism for saliency prediction extends to other domains?
>
> **R**: The key components of our framework, such as feature factorization and probabilistic inference, are general and adaptable, which can be readily extended to other domains and applications. For instance, our method can be directly applied to similar regression tasks in other domains, e.g., visual aesthetic estimation, to gain insights into the contributions of different semantics to the predicted scores. It can also be extended to classification tasks such as image classification, and help identify the common/distinct semantics for different classes. For this, we can adjust our paradigm based on the general methodology discussed in [42], and interpret the relationship between object classes and fine-grained concepts by iteratively analyzing the contributions of semantics for predicting each class.
>
> **3. Q**: It seems that a single epoch of fine-tuning is closer to careful fine-tuning than the original weights. Is this from using a relatively large learning rate? How do the results vary with the learning rate?
>
> **R**: In general, the shift of semantic contributions in saliency prediction models during fine-tuning is larger in the first epoch compared to the subsequent epochs. This is because the pre-trained ImageNet features provide a good initialization of model weights for saliency models to converge quickly. The learning rate used in our experiments is 1e-4, following the learning recipe of the original paper [11]. We also experimented with a learning rate of 4e-4 and observed similar convergence behaviors and results.

---

### Official Review · Reviewer_ZMfD · 2023-07-06

**Soundness:** 3 good
**Presentation:** 2 fair
**Contribution:** 2 fair
**Rating:** 6
**Confidence:** 4

**Summary:**

This paper attempts to decompose the learned representation of a data-driven saliency model into a constituent set of bases that are mapped onto semantic concepts, thereby providing insight into what is driving the model's representation of saliency. This method is applied to three different saliency models of varying formulation over several datasets. Some discussion is then provided of the results, as well as some qualitative discussion of examples of model failures.

**Strengths:**

The paper takes on a challenging and open-ended problem in saliency modelling, namely the difficulty of teasing apart the different contributions to attentional capture. Similarly, the work provides an example of continued work in general explainability within deep learning methods, which is an important issue within the field.

The paper is clearly written and relatively easy to follow, although there are a few details that are missing (such as the choice of N).

**Weaknesses:**

There are some references that seem pertinent that were not discussed. In particular, significant work on failure modes of modern saliency models was poorly represented.
- Bruce et al., "A Deeper Look at Saliency: Feature Contrast, Semantics, and Beyond", CVPR 2016
  -- This paper digs into some of the failure modes common to deep learning models, including object vs. background and semantic vs. feature contrast elements (e.g. see Figure 7), providing pertinent insights to the discussion in the submission.
- Kümmerer et al., "Understanding Low- and High-Level Contributions to Fixation Prediction", ICCV 2017
  -- This paper explicitly explores constituent elements of saliency representation in deep networks from the perspective of high-level vs. low-level features. Given the way the paper attempts to tease apart the representation of saliency between different feature classes, it is conceptually highly relevant background for the current submission.
- Kotseruba et al., "Do Saliency Models Detect Odd-One-Out Targets? New Datasets and Evaluations", BMVC 2019
  -- This paper provides a psychophysical (P^3) and natural image (O^3) dataset with targets explicitly defined by low-level salient features (e.g. colour, orientation, shape, or size singletons), and finds that saliency models (including deep learning-based models) largely perform quite poorly. For exploring failure rates the O^3 dataset would be a potentially useful (albeit ground truth was defined by semantic object annotation and not fixation data), but even if the dataset is not used the examination of model failures in the submission should include the context of this prior exploration.
- Tatler et al., "Visual correlates of fixation selection: effects of scale and time", Vision Research, 2005
  -- This paper explores the evolution of fixations through time, including aspects such as central vs. peripheral distribution and inter-subject consistency of fixation location. Given that this is one area that the submission claims novelty, it would be good to put it in context with prior explorations in this area of the temporal evolution of low-level human attention.

Overall, while the paper is interesting and tackles an exceedingly challenging problem, I think there are a number of conceptual issues that it needs to overcome. While some specific issues are given in the questions below, the primary issue is that while the submission encodes the positive/negative importance of the various bases extracted, it is well established within psychophysics that the relative importance of elements to saliency is contextual (e.g. see Nothdurft (1993) "Saliency effects across dimensions in visual search"; Nothdurft (2000) "Salience from feature contrast: additivity across dimensions" for some low level examples), and so these relative attributes are likely to change from image to image. Within the current submission, these attributes change from model to model and dataset to dataset; what conclusions are to be drawn from this? Is the technique shedding light on dataset composition, model bias, or some deeper aspect of relative aspects of saliency? Much of the analysis is presented without clear connection to either human behaviour (with the exception of Section 4.5, which cleverly makes use of the technique to explore the representations learned from data from different human subject populations or conditions) or model performance in a traditional sense, making it difficult to put into context or derive deeper insight.

**Questions:**

- Although drawing on the architectures of SALICON, DINet, and TranSalNet, the model configurations in this paper are distinct from the instantiations of the original publications in order to accommodate the need for trainable bases. How does that change the ultimate behaviour of the models with respect to standard measures of saliency performance? Alternatively, even a quantified value for the change in saliency maps when compared within model (e.g. the paper's version of SALICON correlated against the standard instantiation of SALICON) would help put the paper's results in context with the existing literature.

- What is the value of N (the number of bases)? How stable are the results with respect to N (i.e. does the mapped semantic content change substantially with even small changes in N)?

- Each basis is mapped onto a top-5 semantic mix. Why not onto a single concept? How was 5 selected? Similar to the previous question, how does this choice affect the subsequent analysis?

- How is "action" defined in a static image? The primary example given, "having meeting", seems like a social activity, which was a separate category. I get that it is challenging to relate the messy details of semantic categorization in a short paper, but given that this is central to the topic of the paper, I think it needs a clearer explanation.

- How do the insights provided in this paper relate to model performance? When the models show a markedly different breakdown of salient factors (e.g. Figure 4, which shows SALICON emphasizing vehicles much more strongly than TranSalNet, while TranSalNet emphasizes clothing more than any other model), does this correlate with predictive accuracy?

- Related to the previous question, could you use the IoU Measurement process used to assign labels to the bases to provide an approximate breakdown of the factors leading to human fixations directly? This might provide another point of comparison to better put the results of this paper in context.

- Given the range of behaviours across the models shown in Figure 4, why does Figure 5 average the semantic weights across models (also, this should be noted in the caption; when I first read the paper I was quite confused which model was being shown in Figure 5)? What is the justification for this? Do the models tend to converge after fine-tuning?

- Figure 6 (b.) and (c.): are these results for subjects without autism? If so, this should be more clearly noted.

- What was meant by line 304: "Whether models... open question."? What would lead to a model behaving "even better than humans", given that humans are the system trying to be modelled?

**Limitations:**

The limitations discussed seem clear.

---

> ### Author Rebuttal · Authors · 2023-08-10
>
> **1. Q**: The relative importance of elements to saliency is contextual. These attributes change from model to model and dataset to dataset; what conclusions are to be drawn from this? Is the technique shedding light on dataset composition, model bias, or some deeper aspect of relative aspects of saliency?
>
> **R**: We agree that the relative importance of elements to saliency is indeed contextual, as demonstrated in the referred psychophysics research on low-level features (Nothdurft, 1993 and 2000) as well as recent ones including higher-level semantics [17, 19, 41]. Instead of investigating individual images, this study aims to reveal general conclusions from learning-based methods based on a large-scale dataset featuring diverse objects in natural contexts. For example, a small far-away pedestrian face in a street image can be less salient than a nearby car, but with many images with faces of different sizes and locations co-occurring with various objects in natural contexts, faces show overall high saliency values. This allows us to make statistical inferences regarding the relative significance of semantics, quantifying and visualizing their contributions to saliency prediction. It further enables a collection of interesting analyses and important conclusions including (1) differentiating the positive and negative contributions of semantics is critical to saliency prediction; (2) deep saliency models learn key properties of attention in different settings (different participant groups, stimuli, and time durations), mostly aligned with findings in human vision studies; and (3) common failures of deep saliency models can be attributed to the inability to differentiate semantics.
>
> The reviewer is right that  “the technique is shedding light on dataset composition, model bias…”. While we highlight the general conclusions, the proposed framework can also be applied to different data or models to reveal certain data- or model-specific conclusions or biases, so readers may be aware of the differences and select them accordingly. For example, Section 4.3 analyzes the same model trained on different datasets and reveals the data characteristics based on semantic weights.
>
> **2. Q**: Some references were not discussed.
>
> **R**: The references are indeed relevant. Our work complements them by differentiating high-level semantics based on their contributions and understanding the general mechanisms of deep saliency models. We will include these papers and add a dedicated section to discuss more comprehensively about failure modes, saliency representation, and temporal evolution in saliency models.
>
> **3. Q**: How does the method change the saliency performance?
>
> **R**: We include the results in the global comment, showing that our method does not compromise saliency performance.
>
> **4. Q**: What is the value of N?
>
> **R**: We set N based on the number of units in the final layers of deep saliency models [9-11]. We explored two settings of N (512 and 1000) and found that their differences in contributions were not substantial. Therefore, we proceed with N=1000, which strikes a balance between granularity and computational efficiency.
>
> **5. Q**: Why not project bases onto a single concept?
>
> **R**: This approach resonates with [42] that uses the top k concepts for interpreting image classification. The top-1 semantic alone may only explain ~30% of the alignment score (sum of normalized IoU). Alternatively, the top-5 semantics account for 90-100% of the score, capturing a broader range of contributing semantics while avoiding an overemphasis on dominant salient/non-salient semantics (e.g., face and cloudness).
>
> **6. Q**: How are semantic categories like action defined?
>
> **R**: Our categorization of semantics considers both their association with objects and meanings. For categories related to actions, we first identify human-centric social semantics (i.e., objects and attributes related to humans), and then distinguish actions (e.g. doing something) from non-actions (e.g., body parts). We also offer detailed results for each semantic in the supplementary materials.
>
> **7. Q**: How do the insights relate to model performance?
>
> **R**:  The paper sheds light on how deep saliency models prioritize visual cues to predict saliency, which is crucial for interpreting model behavior and gaining insights into visual elements most influential in performance: For example, Figure 5 analyzes the evolution of semantic weights through fine-tuning where performance increases with epochs. The fine-tuning improves the weight difference between salient and non-salient semantics, leading to enhanced model performance.
>
> **8. Q**: Could you use the IoU Measurement directly for factorization?
>
> **R**: Directly using IoU for factorization and employing the cosine similarity $\alpha$ as a measure of contribution (which is a form of machine attention) may not quantify the actual contributions of bases (different attention weights lead to the same prediction, see “Attention is not Explanation, 2019”). We address the issue by reformulating saliency prediction with a probabilistic framework, to explicitly measure the contributions and reveal deeper insights into saliency models.
>
> **10. Q**: Are the results in Fig. 6bc for subjects without autism?
>
> **R**: The two subfigures analyze the effects of stimuli and time duration on the observers’ attention, which are unrelated to autism. We will provide clearer explanations to avoid misunderstanding.
>
> **11. Q**: What was meant by line 304?
>
> **R**:  When inter-subject variability is high, human observers do not have a consensus on where to look. In this case, one assumption about saliency modeling (i.e., certain commonality about human attention patterns) may not be true, and the validity of using the ground truth human map for training and evaluation (i.e., the standard leave-one-subject-out approach), and the expected behavior of targeted models are interesting and open questions.

---

### Official Review · Reviewer_u4m5 · 2023-07-07

**Soundness:** 3 good
**Presentation:** 3 good
**Contribution:** 3 good
**Rating:** 7
**Confidence:** 4

**Summary:**

The paper presents a novel analytic framework that provides a principled interpretation and quantification of the implicit features learned by deep saliency models, which are used for predicting human visual attention. The framework decomposes these features into interpretable bases aligned with semantic attributes, and reformulates saliency prediction as a weighted combination of probability maps. The authors conducted extensive analyses to understand the factors contributing to the success of saliency models, including the positive and negative weights of semantics, the impact of training data and architectural designs, and the effects of fine-tuning. They also explored visual attention in various application scenarios, such as autism spectrum disorder, emotion-eliciting stimuli, and attention evolution over time. The study identifies the accurate feature detection and differentiation of semantics as key factors in the models' success, influenced by training data and design choices. The framework is also useful for characterizing human visual attention and understanding common failure patterns in saliency models. The authors suggest incorporating structures and lower-level information for improved modeling. The research has potential impacts on optimizing human-computer interfaces, assisting visually impaired individuals, and enhancing societal benefits.

**Strengths:**

1. The paper presents a novel analytic framework that provides interpretation and quantification of the implicit features learned by deep saliency models for predicting human visual attention. The framework decomposes implicit features into interpretable bases aligned with semantic attributes, allowing for a weighted combination of probability maps connecting the bases and saliency in saliency prediction. The framework effectively identifies a variety of semantics learned by deep saliency models, including social cues, actions, clothing, and salient object categories, showcasing its versatility in analyzing attention across diverse scenarios.
2. The framework reveals the positive and negative contributions of semantics to saliency, highlighting the ability of deep saliency models to distinguish between salient and non-salient semantics.
3. The analysis demonstrates how training data and model designs impact saliency prediction, with shifts in semantic weights reflecting the characteristics of the datasets and models.
4. The study also investigates the effects of fine-tuning on semantic weights, showing how deep saliency models progressively adapt features during training to better capture salient cues and refine the weights of negative semantics.
5. The framework is also applied to explore the capture of human attention characteristics, such as the impact of visual preferences, characteristics of visual stimuli (e.g., emotions), and temporal dynamics, providing insights into the factors influencing attention deployment.
6. The findings validate the effectiveness of deep saliency models in automatically identifying salient semantics, differentiating foreground from background, and encoding fine-grained characteristics of attention.

**Weaknesses:**

The work has a lot of novelty with good empirical analyses. The paper is also nicely written. However, I find that the authors have missed comparing the performance of their model with other saliency models using standard saliency metrics as can be found in this leaderboard: http://saliency.mit.edu/results_mit300.html.

The authors have also failed to cite some important works from like DeepGaze, DeepFix, BMS, etc.

**Questions:**

I am curious to know if the authors could perform and share the results of comparison with existing methods using the standard metrics for measuring saliency.

**Limitations:**

The authors analyzed the failure patterns of deep saliency models within the intermediate inference process using their proposed factorization framework. They select common success and failure examples where three tested models consistently have high/low NSS scores. They then perform a qualitative analysis by visualizing the spatial probabilistic distribution of the bases for semantics with positive and negative weights.

In the successful examples, they find that accurate saliency prediction is correlated with the differentiation of diverse semantics. The stimuli in these examples typically have salient and non-salient regions belonging to different semantics. Therefore, the models, with the ability to distinguish positive and negative semantics, can readily determine the saliency distribution. On the other hand, in the failure examples, the models struggle to determine saliency within objects or among objects with similar semantics. Investigation of the probabilistic distribution of bases reveals that models often have a uniform-like distribution of bases on object parts or among objects of the same category, making it difficult to construct accurate saliency maps.

The analysis also highlights that existing models have difficulty with scenes without salient objects. These scenes are challenging for the models as they lack clear focal points. The ground truth human attention in these failure patterns exhibits high inter-subject variability, suggesting that human viewers may not agree on where to look, making the ground truth maps less reliable. The authors raise the question of whether models may perform as well as or even better than humans in these challenging situations.

Based on their observations, the authors hypothesize that leveraging more structured representations to encode contextual relationships between semantics and integrating mid- and low-level cues may be beneficial in addressing these failure patterns in certain scenarios. This suggests that incorporating additional information and incorporating contextual understanding may help improve the performance of deep saliency models.

---

> ### Author Rebuttal · Authors · 2023-08-10
>
> **1. Q**: The work has a lot of novelty with good empirical analyses, but has missed comparing the performance of their model.
>
> **R**:  We acknowledge the importance of performance comparison with existing methods using standard metrics for measuring saliency, and have added comparisons in the global response. Results show that our approach is able to achieve competitive performance among three popular datasets. We are committed to including these comparisons and results in the revised version of the paper.
>
> **2. Q**: Cite some important works from like DeepGaze, DeepFix, BMS, etc.
>
> **R**: We appreciate the suggestion to highlight relevant works. These references will certainly enrich our discussion on diverse saliency techniques and we will add them in the revision.

---

> > ### Comment · Reviewer_u4m5 · 2023-08-17
> >
> > I appreciate the authors' efforts to respond to all concerns raised by the reviewers. I would like to keep my rating.

---

### Author Rebuttal · Authors · 2023-08-10

We thank the reviewers for their insightful feedback. We are encouraged that they recognize our work as solving a fundamental, important, challenging, and open-ended problem, which approaches explainability and takes a step towards truly understanding the success of deep learning models (ZMfD, srbe, UKcT). We appreciate that they find our work with a lot of novelty and good empirical analysis, on multiple aspects including both salient and non-salient semantics, data and model impact, and the exploration of human attention characteristics (u4m5, ZMfD, srbe, UKcT). All reviewers identify the importance of feature decomposition and explicit modeling of the connections between bases and semantics, indeed two key components of our framework for bridging the gap between visual semantics and saliency. We are also glad that they consider our paper nicely written (u4m5, ZMfD, UKcT).

The key objective of our study is to develop a principled framework for interpreting and understanding the underlying mechanism behind deep saliency models, rather than designing saliency models that strive for enhanced performance. As pointed out by Reviewer srbe, instead of competing to marginally beat an ROC score, it approaches saliency from a different point of view and works toward a true understanding of deep saliency models. With this objective in mind, by explicitly quantifying the contributions of diverse semantics, our approach provides key insights into the impacts of different factors (e.g., datasets, model architecture, learning process in Section 4.2-4.4) on saliency prediction and attention deployments in various settings (e.g., different participant groups, stimuli with diverse sentiment, and varying temporal dynamics in Section 4.5) and also complements studies on model design [9, 10, 11] with an interpretable tool for investigating the behaviors of models (e.g., Section 4.6).

Although not the focused point of the paper as acknowledged by the reviewers, our approach also shows promise in achieving competitive performance in saliency prediction. To complement our analyses in the main paper and further substantiate the efficacy of our methodology in accurately interpreting saliency models without altering their inherent behaviors, we follow reviewers' suggestions and include an additional performance comparison. The table below shows the comparative performance across three commonly used datasets: OSIE, MIT, and SALICON (our model is trained using the SALICON training split with a DINet backbone). These results demonstrate the competitive nature of our approach compared with state-of-the-art techniques. It is noteworthy that our approach introduces minimal architectural modifications, limited to the last two layers of the saliency models (see Section 4.1 for details), thereby ensuring that its performance aligns seamlessly with the original DINet model across all datasets.


| | | CC | NSS |
|----------|----------|----------|----------|
| OSIE  | SALICON| 0.63 | 2.75 |
| | SAM | 0.65 | 2.70 |
| | DINet| 0.63 | 2.88 |
| |  Ours (DINet) | 0.64 | 2.91 |
| MIT | DVA| 0.64 | 2.38 |
| | SALICON | 0.70 | 2.56 |
| | SAM| 0.69 | 2.47  |
| | DINet | 0.70 | 2.54 |
| |  Ours (DINet)  | 0.70 | 2.53 |
| SALICON | DeepNet| 0.86 | 1.61 |
| | SAM | 0.86 | 1.84 |
| | SALICON | 0.86 | 1.89 |
| | UNISAL | 0.88 | 1.95 |
|  | EML-Net | 0.87 | 1.95 |
| | DINet  | 0.87 | 1.92 |
|    | Ours (DINet) | 0.86 | 1.89 |

We address the questions (**Q**) raised by each reviewer in the individual responses (**R**) below, and will incorporate the comments in the revision.

---

> ### Author Response · Authors · 2023-08-16
>
> We appreciate the reviewers for their valuable comments! Please let us know if there are any further feedbacks and we are more than happy to discuss and improve our paper.

---

### Decision · Program_Chairs · 2023-09-21

**Decision:**

Accept (poster)

**Comment:**

Paper received 1 x Accept, 2 x Weak Accept and 1 x Borderline Reject ratings. A number of concerns were raised in the initial reviews. However, provided rebuttal largely addressed concerns of [ZMfD] and [u4m5] as noted in their post-rebuttal discussion; [srbe] did not participate in discussion but was already positive about the paper pre-rebuttal. Reviewer [UKcT] engaged in lengthy discussion with the authors but found the responses only partially convincing at the end. Remaining concerns of [UKcT] were carefully reviewed and discussed by AC and SAC. While valid, they were deemed not significant enough to prevent acceptance of the paper. Ultimately, AC agrees with the consensus that paper presents interesting and valuable findings and would be a valuable addition to the conference. Authors are encouraged to revise the manuscript to include discussion and additional results provided in the rebuttal.